# Dimensionality, information and learning in prefrontal cortex

**Ramon Bartolo**[ID]**, Richard C. Saunders, Andrew R. Mitz**[ID]**, Bruno B. Averbeck**[ID]*

Laboratory of Neuropsychology, National Institute of Mental Health, National Institutes of Health, Bethesda, Maryland, United States of America

* bruno.averbeck@nih.gov

## Abstract

Learning leads to changes in population patterns of neural activity. In this study we wanted to examine how these changes in patterns of activity affect the dimensionality of neural responses and information about choices. We addressed these questions by carrying out high channel count recordings in dorsal-lateral prefrontal cortex (dlPFC; 768 electrodes) while monkeys performed a two-armed bandit reinforcement learning task. The high channel count recordings allowed us to study population coding while monkeys learned choices between actions or objects. We found that the dimensionality of neural population activity was higher across blocks in which animals learned the values of novel pairs of objects, than across blocks in which they learned the values of actions. The increase in dimensionality with learning in object blocks was related to less shared information across blocks, and therefore patterns of neural activity that were less similar, when compared to learning in action blocks. Furthermore, these differences emerged with learning, and were not a simple function of the choice of a visual image or action. Therefore, learning the values of novel objects increases the dimensionality of neural representations in dlPFC.

## Author summary

In this study we found that learning to choose rewarding objects increased the diversity of patterns of activity, measured as the dimensionality of the response, observed in dorsal-lateral prefrontal cortex. The dimensionality increase for learning to choose rewarding objects was larger than the dimensionality increase for learning to choose rewarding actions. The dimensionality increase was not a simple function of the diverse set of images used, as the patterns of activity only appeared after learning.

## Introduction

Behavior is driven by activity in populations of neurons [1]. The neural activity patterns in populations are high dimensional measurements. However, in some cases, high dimensional population activity may reside on a lower-dimensional manifold and therefore activity can be described well after projection into a low-dimensional subspace [2, 3]. When neural activity is

**Data Availability Statement:** All data and code is available from https://data.mendeley.com/datasets/p7ft2bvphx/draft?a=7a6944f1-0173-422f-abb1-caa2169d2333.

**Funding:** This work was supported by the Intramural Research Program of the National Institute of Mental Health (ZIA MH002928 (BA)). The funders had no role in study design, data collection and analysis, decision to publish, or preparation of the manuscript.

**Competing interests:** The authors have declared that no competing interests exist.

low dimensional, the population only explores a subset of the dimensions that it theoretically could, given the number of neurons in the population. The relevant dimensions can be assessed by finding a set of basis vectors, or population activity patterns, that can be used to accurately reconstruct population activity.

The dimensionality of neural responses is often assessed by first computing the average response of each neuron, in each task condition, and then characterizing the dimensionality of these average responses [4, 5]. Because dimensionality is usually characterized on the mean activity of the population across different task conditions, it is a function of the single neuron tuning functions across time and across task variables [6]. If single neurons have strong temporal autocorrelations during tasks, the neural activity of the population does not change quickly over time, and therefore the population does not explore multiple dimensions in time, within a condition. Similarly, if neurons are broadly tuned to the task parameters being studied, population activity will change slowly as visual inputs or motor responses vary. Dimensionality, therefore, characterizes the diversity of activity patterns, generated in a population of neurons, across task conditions.

Information, like dimensionality, also characterizes activity patterns across task conditions. Linear information in a population of neurons, including linear Fisher Information, is a measure of the signal to noise ratio [7, 8]. The signal is a measure of the difference between patterns of activity between conditions. If mean population patterns of activity are similar for two different task conditions, the population will have minimal information about those conditions. The noise is a measure of the variability in the population, within a condition. The noise that is relevant to linear information is the noise in the dimensions that differ across conditions [9, 10]. If there is minimal information, the conditions cannot be reliably decoded. Information, therefore, is also a measure of differences in activity patterns across conditions. Information, however, unlike most dimensionality measures, also takes into account the amount of noise in the task dimensions.

During learning, behavior and patterns of activity in the brain change [11–16]. Because both dimensionality and information depend on patterns of activity, this raises the question of how learning affects these measures, and how the effects may be related. To address these questions, as well as the question of how neural coding may underlie important learning related behaviors, we trained animals on a two-armed bandit RL task, in which animals learned to associate rewards with choices of visual stimuli or choices of actions [17]. While the animals carried out the task we recorded population activity using 8 Utah arrays implanted bilaterally in area 46 of dorsal-lateral prefrontal cortex (dlPFC)[18]. We found that learning to associate novel pairs of objects with rewards expanded the dimensionality of population representations in dlPFC and therefore drove activity into novel regions of population coding space. However, learning to associate the same actions with rewards, in different blocks of trials, occurred in mostly overlapping dimensions of population activity.

## Results

### Task and behavior

We trained two macaques on a two-armed bandit reinforcement learning task (Fig 1A). The task was run in 80 trial blocks, and at the beginning of each block two new images were introduced that the animal had never seen before. In addition, the task had two conditions (Fig 1B). In the What condition the animals had to learn which of the two images, randomly presented left and right of fixation, was more frequently rewarded (reward probability = 0.7 vs 0.3), independent of its location. In the Where condition they had to learn which of two saccade directions was more frequently rewarded, independent of the image that was chosen. The condition

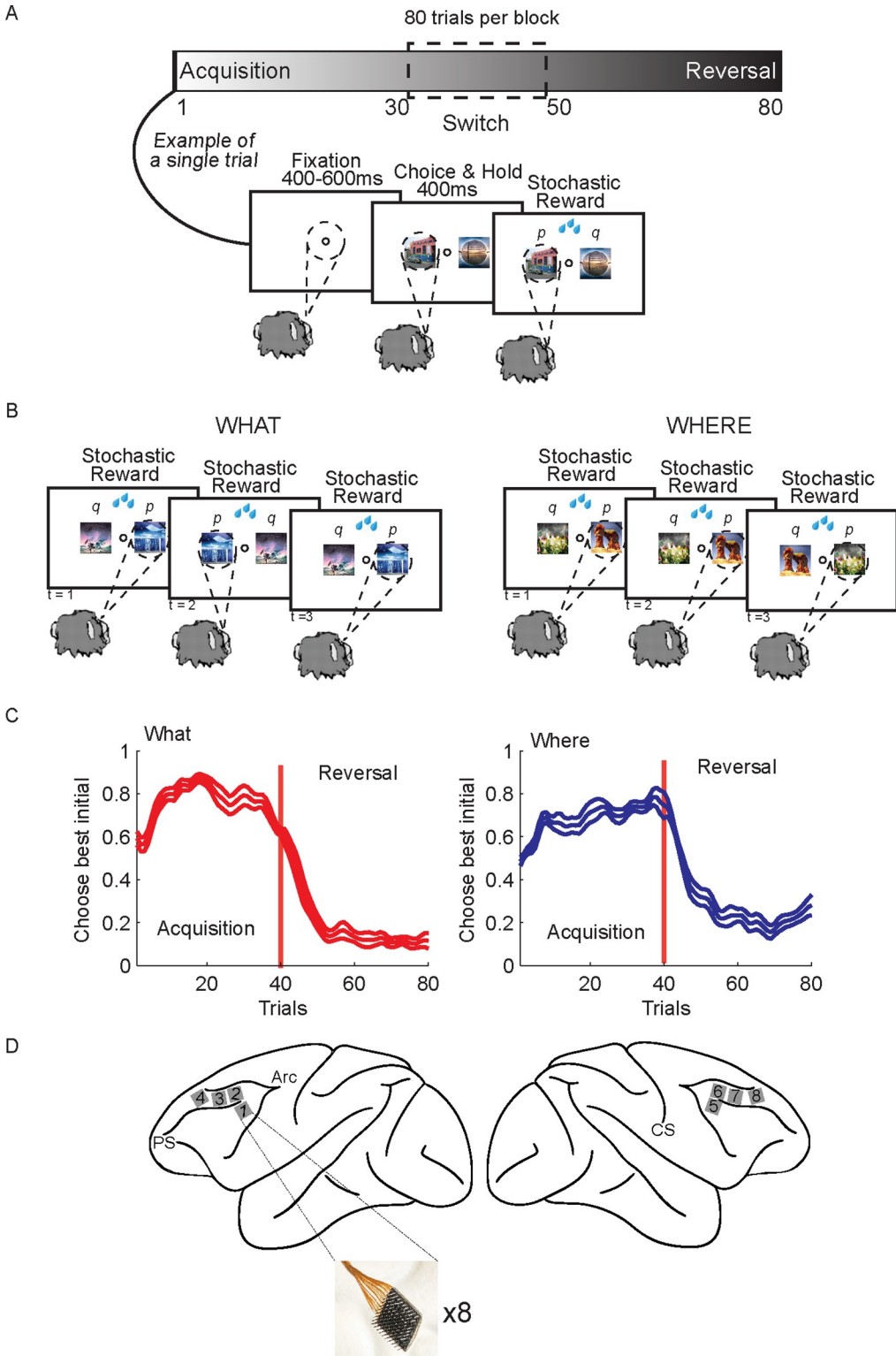

**Fig 1. Two-armed bandit reinforcement learning task, behavior and recording locations.** A. The task was carried out in 80 trial blocks. At the beginning of each block of trials, 2 new images were introduced that the animal had not seen before. In each trial the animals fixated, and then two images were presented. The images were randomly presented left and right of fixation. Monkeys made a saccade to indicate their choice and then they were stochastically rewarded. B. There were two conditions. In the What condition one of the images was more frequently rewarded

(p = 0.7) independent of which side it appeared on, and one of the images was less frequently rewarded (p = 0.3). In the Where condition one of the saccade directions was more frequently rewarded (p = 0.7) and one was less frequently rewarded (p = 0.3) independent of which image was at the chosen location. The condition remained fixed for the entire block. However, on a randomly chosen trial between 30 and 50, the reward mapping was reversed and the less frequently chosen object or location became more frequently rewarded, and vis-versa. C. Choice behavior across sessions. Animals quickly learned the more frequently rewarded image (left panel) or direction (right panel), and reversed their preferences when the choice-outcome mapping reversed. Because the number of trials in the acquisition and reversal phase differed across blocks, the trials were interpolated in each block to make all phases of equal length before averaging. The choice data was also smoothed using Gaussian kernel regression (kernel width sd = 1 trial). Thin lines indicate s.e.m. across sessions (n = 6 of each condition). D. Schematic shows locations of recording arrays, 4 in each hemisphere. Array locations were highly similar across animals.

remained fixed within a block and blocks of each condition were randomly interleaved. The learning condition was not cued and had to be inferred from the rewarded outcomes. In both conditions the choice-outcome mapping within a block was reversed on a randomly chosen trial between 30 and 50, such that the more frequently rewarded choice became less frequently rewarded, and vice-versa, always staying within the same condition. The animals were able to quickly identify the condition, as well as the more frequently rewarded image or direction, and reverse their preferences when the choice-outcome mapping reversed (Fig 1C). Because each block began with new images, we could study the learning process repeatedly. While animals carried out the task, we recorded neural activity using 8 Utah arrays implanted bilaterally in dorsal and ventral area 46 (dlPFC; Fig 1D). The arrays allowed us to record up to 1000 neurons simultaneously (N = 585, 747, 677, 1026, 877, 598 in 6 analyzed sessions, 3 from each monkey).

## Single neuron analyses

Previous work has shown that single neurons in dlPFC represent important aspects of learning and choice behavior [13, 15, 19–22]. Consistent with this, we found single neurons that responded to chosen objects and chosen directions (e.g. Fig 2A–2F). When trials of specific conditions were aligned for single cells, the task features to which the neurons were responsive were often clear. For example, neurons often fired more strongly during the selection of one object vs. the other in each block (Fig 2E and 2F) and also fired more strongly to the choice of one direction vs. the other (Fig 2E and 2F). Across the population, the information relevant to the task was well-represented at the single neuron level (Fig 2G and 2H). The animals learned the best objects and values in each block, and therefore the representation of the chosen object, chosen direction and their values were elevated during the ITI and baseline fixation. This reflects the learning. The reward outcome, however, could not be predicted until it was delivered, and therefore it was at chance levels until delivery. This analysis shows that the task engaged a large fraction of the population of neurons from which we recorded.

## Population dimensionality

The activity of a population of neurons, in a single trial within a time window, can be thought of as a high dimensional vector (Fig 3A and 3B). Although measured neural responses in a population are high dimensional, recent work, has shown that task related neural activity is often low dimensional [2, 23, 24]. In other words, neural activity often exists in a small number of dimensions, relative to the size of the recorded population. Therefore, we sought to characterize dimensionality in our data and relate dimensionality to information about choices in single trials.

To characterize dimensionality in our data we calculated the mean activity vector, $\mu$, for each choice (i.e. object 1 vs. object 2 in What blocks, left vs. right in Where blocks). We did

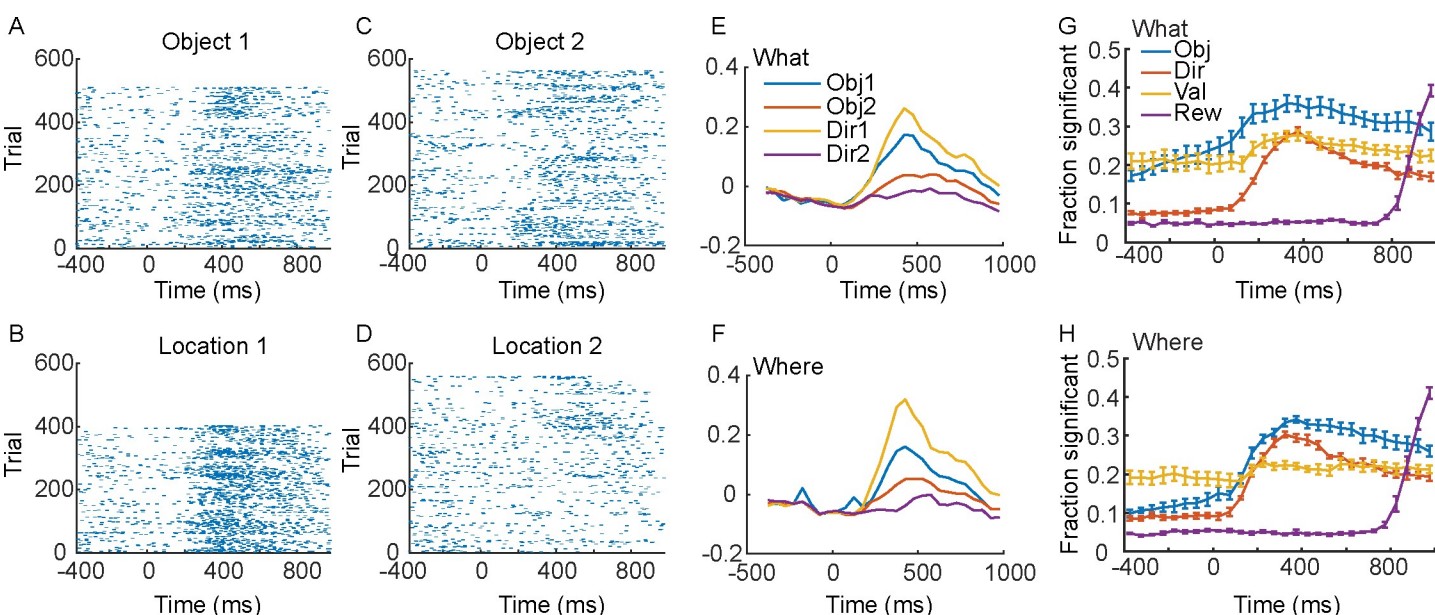

**Fig 2. Single cell example and single cell population statistics.** Time 0 is cue onset. Rasters and spike density functions for the example neuron are in panels A-F. A. Responses to choosing Object 1 in the What condition. B. Responses to direction/location 1 in the Where condition. C. Responses to choosing Object 2 in the What condition. D. Responses to choosing direction/location 2 in the Where condition. E. Average responses for the single neuron shown in A-D in the What condition. Average of the spike density functions for the preferred (Obj 1) vs. non-preferred (Obj 2) objects, or direction 1 or 2. F. Same as E for the Where condition. G. Fraction of neurons across the population significant in an ANOVA for each factor indicated, in the What condition. H. Same as G for the Where condition. Solid lines are averages across sessions (N = 6, 3 from each animal) and error bars are s.e.m. across sessions.

this in each block, separately for the acquisition and reversal phases. The length of this vector was equal to the size of the simultaneously recorded population. The mean activity (i.e. trial averaged spike count) was further estimated in two, 250 ms bins, time locked to cue onset (Fig 3A; $x_1$ = 1–250 ms and $x_2$ = 251–500 ms) for each neuron. We collected these mean estimates for all neurons into vectors, $\mu$, and collected the vectors into a matrix, $U = [\mu_{1,1,1,1} \ldots \mu_{l,m,n,o} \ldots \mu_{2,2,2,24}]$. The matrix $U$ had 192 columns because each column of this matrix was one mean response vector $\mu$ (where the indices take on values $l$ = 2 time bins x $m$ = 2 choices x $n$ = 2 phases x $o$ = 24 blocks; see methods). The number of rows in $U$ was given by the number of neurons simultaneously recorded. The maximum dimensionality in our data was, therefore, 192, because of the number of conditions analyzed. This dimensionality is the same if one extracts eigenvectors from the Neuron x Neuron covariance matrix or the Conditions x Conditions covariance matrix [6]. We then asked whether the activity from all the blocks could be spanned by a smaller set of vectors [2]. If the mean activity vectors lie in a low dimensional space, information about choices in our task would lie in this low dimensional space. We examined this by carrying out singular value decomposition on the matrix $U$. We found that we needed 25 dimensions to account for 80% of the variance and 52 dimensions to account for 90% of the variance in the activity (Fig 3C). Therefore, the activity across the blocks did not lie in a space spanned by a few dimensions. Dimensionality was not, however, maximal, where the maximum dimensionality would be 192.

Next, we examined dimensionality in What and Where blocks separately (Fig 3E). When we did this we found that there was considerable overlap, but dimensionality was higher in What blocks than in Where blocks (F(1, 5) = 55.3, p < 0.001). We also estimated the number of dimensions required to account for 80% of the variance as a function of the number of blocks included in the analysis, by randomly combining subsets of blocks. We found that the

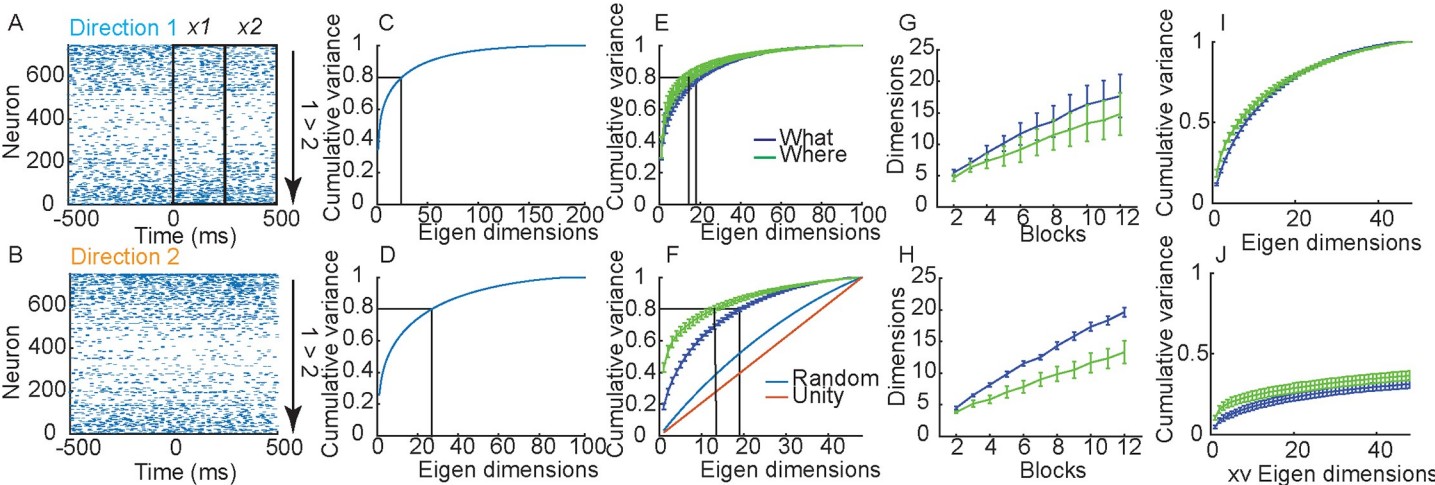

**Fig 3. Single trial representations.** A. Raster showing the response of a population of simultaneously recorded neurons from a single trial when the animal chose Direction 1 in a trial from the Where condition. Outline boxes show time windows, $x_1$ and $x_2$, used to define activity vectors in analyses. B. Same as A for choice of Direction 2. C. Cumulative fraction of variance explained in trial-averaged responses (i.e. the vectors $\mu_i$ from each block and phase) across all blocks of both conditions. Note that the matrix only has 192 independent dimensions, so the cumulative variance saturates at 1 in dimension 192. D. Same as C for information dimensions, $w = \mu_2 - \mu_1$. This matrix has 96 dimensions, so cumulative variance saturates in dimension 96. E. Cumulative fraction of variance explained in trial-averaged responses (i.e. the vectors $\mu_i$ from each block and phase). Data is split out by What (Blue) and Where (Green) blocks. F. Same as E for informative dimensions, $w$. In addition, we also plot the dimensionality of a matrix with the same dimensionality, but with vectors chosen to have random directions, and also the unity line which has a constant increase in variance. G. Dimensionality of trial averaged responses as a function of number of blocks included in analysis. As we accumulate blocks in the analysis the dimensionality increases, also showing that the means do not all lie in the same subspace. The y-axis was estimated by calculating the number of dimensions required to account for 80% of the variance in the PSTHs as we aggregated randomly selected blocks. Note that the more independent (orthogonal) are the PSTHs in different blocks, the more dimensions will be required to span them as we aggregate across blocks. H. Same as G for the informative dimensions. I. Cumulative variance accounted for in informative dimensions, $w$, after removing the first principal component from both What and Where conditions. J. Cumulative variance accounted for with cross validation.

dimensionality expanded in both What and Where conditions as we included more blocks, but did so more quickly for What blocks than Where blocks (Fig 3G; $F(1, 5) = 59.4$, $p < 0.001$).

The previous analyses characterized the dimensionality in the mean responses for each choice, $\mu_i$. We next examined the dimensionality of the informative dimensions. For large populations, ignoring noise correlations (see methods), the informative linear dimension in a single block is given by $w = \mu_{m=2} - \mu_{m=1}$, i.e. the difference in the mean response vectors for the two choices [25]. Therefore, we calculated the informative dimension for each block and phase, by taking the difference between the mean response vectors. Then we collected the $w$'s from each block, phase and time bin into a large matrix $W = [w_{1,1,1} \dots w_{l,n,o} \dots w_{2,2,24}]$, as we had done for the mean responses. This matrix had 96 columns because we have collapsed the two choice vectors $\mu_1$ and $\mu_2$ into a single difference vector, $w$ and therefore we halved the number of dimensions. When we carried out SVD on this matrix we found that we needed 28 dimensions to capture 80% of the variance (Fig 3D). When we examined the task conditions separately, we found that dimensionality was higher in What blocks than in Where blocks (Fig 3F; $F(1, 5) = 54.82$, $p < 0.001$) and the dimensionality grew more quickly in What blocks than Where blocks (Fig 3H; $F(1, 5) = 21.4$, $p = 0.006$). In What blocks we needed 20 dimensions to account for at least 80% of the variance in the informative dimensions and in Where blocks we needed 14. Most of the difference between conditions, however, was driven by the first principal component. When this was removed in both conditions, there was no difference between conditions (Fig 3I; $F(1, 5) = 4.33$, $p = 0.092$).

We also examined the variance accounted for in cross validated data. We removed one trial of each choice (e.g. one trial of choosing object 1 and one trial of choosing object 2), estimated the SVD using the remaining trials, and then projected the held-out trial onto the eigen

dimensions (Fig 3J). Note that the eigen-dimensions in these analyses (Fig 3) were computed on the mean activity vectors from each block and phase, and when we cross-validate we are computing the variance accounted for in a single trial, as opposed to the mean across trials. Therefore, much of the variance of the single trial will be due to noise. Thus, the cross-validation analysis only captured a small amount of the variance in the held-out data. However, the What condition was still higher dimensional than the Where condition ($F(1, 5) = 15.2$, $p = 0.012$).

The difference between conditions was more pronounced in the informative dimensions (Fig 3F; $w = \mu_2 - \mu_1$) than it was in the mean activity (Fig 3E, $\mu_i$). The reason is that the vector $w$ removes any components that are common to $\mu_1$ and $\mu_2$. Specifically, if $\mu_i = x_i + c$, where $x_i$ is specific to choice $i$ and $c$ is common to both choices, the difference will only depend on $x_i$: $w = x_2 - x_1$. The similarity in dimensionality between the two conditions seen in the mean activity (Fig 3E) followed from the presence of common components, $c$, that had high variance. These components reflect aspects of behavior that are common to all task conditions. (Note that if we subtract the average activity across the two choices $c = (\mu_2 + \mu_1)/2$ from the mean activity in each block, $x_i = \mu_i - c$, and examine the dimensionality of the $x_i$, it is identical to the dimensionality of the $w$'s.) The subsequent analyses will be based upon the $w$'s as these reflect the dimensionality relevant to the chosen options in each block. They can also be linked directly to population information.

The previous studies that found low-dimensional representations used overlearned tasks with few conditions. Although our task had only a few conditions, our conditions were not conditions under which choices were made, they were conditions under which the animals learned to make choices between objects or directions that differed in value. This learning may have led to the dimensionality expansion across blocks we found in our task. More specifically, the dimensionality in our task might be driven by learning preferences over new sets of objects or direction in each block. This learning might drive representations into new, independent regions or subspaces, in dlPFC. The learned representation in a single block might be low dimensional, but every time choices were learned in a new block, activity was driven into a new region of coding space, and correspondingly a new low dimensional subspace in dlPFC. Therefore, across the experiment, the dimensionality was not low because every time a new pair of images was learned, the dimensionality expanded. Every time the values of a new pair of objects were learned the brain generated new patterns of activity. In the next analyses we further explored this hypothesis.

## Information and decoding

The dimensionality of the matrix $W$ is a measure of the extent to which the informative dimensions, $w$, from all blocks can be spanned by a smaller number of dimensions. This is a measure of how orthogonal (or not) the $w$'s are. To further characterize this, we examined the extent to which the informative dimensions were shared or similar across blocks of trials. If the informative dimensions from two blocks, A and B, are similar (i.e. the angle between them is small), they can be used interchangeably for decoding. However, if they are close to orthogonal, there will be little information about choices in block A, when the $w$ from block B is used for decoding.

For an arbitrary discriminant line, $w_s$, the information about choices in a block of trials, where the means for the two options are given by $\mu_1, \mu_2$ is: $I = \frac{(w_s^T(\mu_2 - \mu_1))^2}{w_s^T Q w_s}$. This equation shows that the angle between $w_s$ and $w = \Delta\mu = \mu_2 - \mu_1$, in the numerator, strongly affects the information. If $w_s$ and $\Delta\mu$ are orthogonal, information will be zero. If $w_s$ is the discriminant line from a different block of trials, then the shared information across blocks will be a measure of the

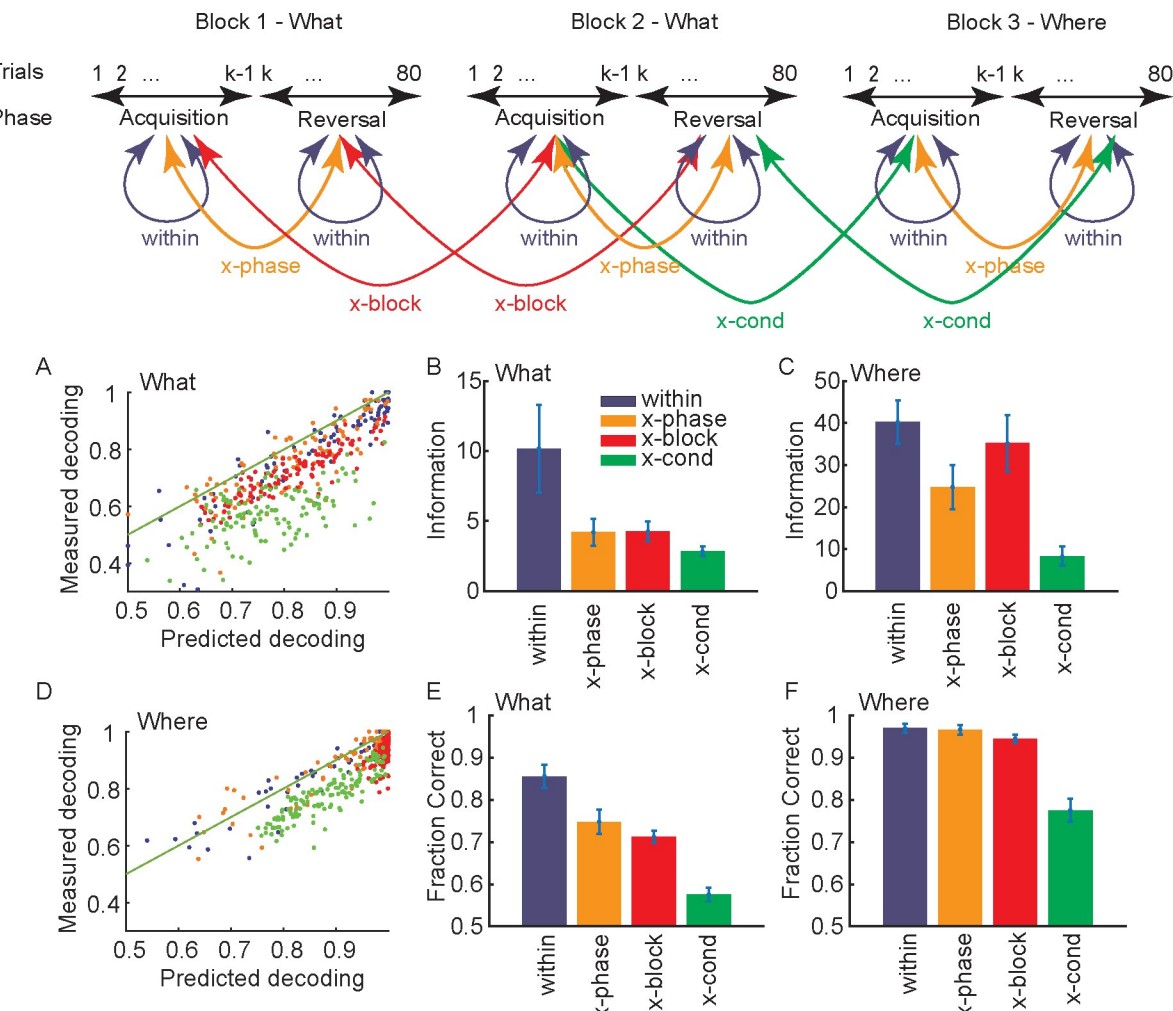

**Fig 4. Information in subspace defined in various ways.** Error bars in bar plots are s.e.m. and n = 6 (i.e. sessions) in all cases. Diagram at the top shows example of how comparisons are defined. A. Relation between predicted and measured decoding performance for What condition. Units are fraction correct. B. Information in the What condition in subspaces defined for the current block and phase (within), for the opposite phase from the same block (x-phase), for the same phase for other blocks of the same type (x-block) and for the same phase for other blocks of the other type (x-cond). C. Same as B for the Where condition. D. Relation between predicted and measured decoding performance for Where condition. Units are fraction correct. E. Decoding performance shown as fraction correct for the What condition. F. Same as E for the Where condition.

angle between $w_s$ and the $w$ from the current block, since $w = \mu_2 - \mu_1$. Although $w$ is 2xN dimensional for each block and phase, because of the two time windows, it can be vectorized for calculations (see methods).

To examine this, we calculated information about choices in a block using various subspaces (i.e. $w$'s from different phases or different blocks) identified in 4 ways (Fig 4 top). First, we used the $w$ that corresponded to the informative dimensions from the corresponding block and phase (within). Second, we used the $w$ for the opposite phase (i.e. acquisition to reversal and vis-versa) from the same block (x-phase) to calculate information about activity in the current phase. Third, we used $w$'s identified for the other blocks of the same condition and phase (x-block). Fourth, we used $w$'s identified for the same phase of the blocks of the other condition (x-cond). If dlPFC used a (relatively) independent representation for each pair of objects learned, there should be (significantly) less information about choices, when $w$'s from other

blocks of either the same or opposite condition were used. If learning the values of a new pair of objects drove activity into new dimensions in dlPFC, then there should be relatively little information about choices shared across the relevant dimensions from different blocks. In contrast to this, choices may be represented relative to value, in which case they may re-use the same subspace, similar to what has been seen for BCI learning on short [26], but not long time scales [27]. Furthermore, by comparing information between the acquisition and reversal phases, we could see if changing choice-outcome associations drove activity into new dimensions, when both the visual input and motor output remained constant, but the values reversed. In addition, our design allowed us to compare the change across blocks of the What and Where conditions. Because the saccade directions to be learned were preserved across blocks, the Where condition controlled for confounding factors like time within a recording session and potential non-stationarity in neural responses. If information was reduced across blocks because of drifting neural activity, it should be present in Where blocks as well as What blocks.

We first characterized similarity between $w$'s after learning, during the performance phase. We did this by computing both information and decoding performance (see methods), in the trials after learning (i.e. $\geq$ trial 5 in each block) using leave 3 out cross-validation, where we left out the trial to be tested as well as the preceding and following trials (Fig 4). For all analyses, we projected the single-trial data from a given block (and phase) into one of the subspaces, and then re-estimated the decoding boundary after the projection. Therefore, if the neural activity only re-organized within the same dimensions, we would not lose information. However, if the neural activity re-oriented into different dimensions, it would no longer be discriminable.

For the What condition, information and fraction correct varied depending on which subspace (i.e. $w$) was used (Fig 4B and 4E, Information: $F(3, 15) = 18.3$, $p < 0.001$; Decoding: $F(3, 16) = 66.9$, $p < 0.001$). Information and accuracy were higher, within than across phases (Information $F(1, 5) = 17.3$, $p = 0.012$; Decoding: $F(1, 2) = 88.6$, $p = 0.009$), between blocks of the same condition (Information: $F(1, 15) = 14.9$, $p = 0.013$; Decoding: $F(1, 15) = 52.7$, $p = 0.001$) and were lowest in subspaces identified for the Where condition when they were used on activity from the What condition (Information: $F(1, 15) = 16.9$, $p = 0.010$; Decoding: $F(1, 16) = 208.9$, $p < 0.001$). Therefore, learning values associated with new pairs of objects occurred in relatively independent subspaces. Furthermore, reversing the stimulus-outcome mapping also drove activity into a new subspace, and the largest difference in subspaces was across conditions. There was, however, also some shared information across subspaces. Neither information nor decoding performance were driven to 0 or chance levels respectively.

Our primary focus, however, was on differences across conditions. Therefore, we next examined decoding accuracy and information in the Where condition. We found that there was a difference across the 4 subspaces (Fig 4C and 4F: Information: $F(3, 15) = 32.9$, $p < 0.001$; Decoding: $F(3, 15) = 32.9$, $p < 0.001$). However, there was no difference between phases in decoding ($F(1, 4) = 0.1$, $p = 0.744$) but there was in information ($F(1,4) = 25.7$, $p = 0.006$). There was also no difference in decoding between blocks of the same condition ($F(1, 4) = 6.0$, $p = 0.068$) but there was in information ($F(1, 5) = 11.6$, $p = 0.022$). There was lower decoding performance and information in subspaces identified for What blocks when they were used on activity from Where blocks (Information: $F(1, 5) = 75.9$, $p < 0.001$; Decoding: $F(1, 5) = 30.0$, $p = 0.003$). Therefore, decoding of the chosen direction tended to be preserved across blocks of the same condition and phases within blocks. However, there was a decrease in information. When we directly compared conditions, we found that different blocks of the What condition were more independent than blocks of the Where condition, because information dropped more between the Within subspace and the other conditions for What blocks than Where

blocks (Blocktype x Subspace, Information: $F(3, 536) = 21.6$, $p < 0.001$; Decoding: $F(3, 507) = 18.3$, $p < 0.001$).

On average, decoding and information were correlated (Fig 4A and 4D, What: $r = 0.837$, $p < 0.001$; Where: $r = 0.865$, $p < 0.001$), although, measured decoding tended to be below predicted decoding, particularly for the x-cond comparison. To some extent the over-estimation of actual decoding is due to cross validation and the limited number for trials. Decoding and information can also diverge, particularly with limited trials, if population responses remain separated in the subspace, but move closer together (e.g. see below Fig 6). Decoding and information are a function of signal and noise in the relevant dimensions. Therefore, we also examined these separately, and found that the difference in decoding performance between the What and Where conditions was due to a change in signal (Fig 5A and 5B; Comparison between the "within" measure across conditions, $t(10) = 2.6$, $p = 0.026$), not a change in noise (Fig 5C and 5D; $t(10) = 0.6$, $p = 0.554$). In an additional analysis we also characterized the principal angle between subspaces, to characterize shared information across blocks. We found that the angle between subspaces was larger for What blocks than Where blocks (Fig 5E and 5F; t-test between x-block of What and Where; $t(10) = 5.9$, $p < 0.001$).

These results support a consistent conclusion. Dimensionality is higher across blocks of the What condition than the Where condition (Fig 3F). There is less shared information (although not 0), and lower shared decoding (although not chance), across phases and blocks of the What condition than the Where condition (Fig 4), which is consistent with the subspaces, $w$, for different blocks being closer to orthogonal for the What condition. And, finally, direct measures of the principal angle between subspaces for blocks of the What condition are higher than between blocks of the Where condition (Fig 5E and 5F). Therefore, dimensionality was higher across blocks of the What condition than across blocks of the Where condition. It is important to note, however, that in neither condition were the dimensions completely orthogonal, and there was some shared information. Therefore, there are some dimensions that carry information about choices across blocks in both the What and Where conditions. Further, we have carried out the information analyses using the discriminant lines and one might wonder what can be inferred about changes in activity, when there are changes in informative dimensions. For example, one might think that the activity was discriminable along discriminant line 1 in block A, and becomes discriminable along discriminant line 2 in block B (as we have shown). And that it is the case that the activity was already present along discriminant line 2 in block A, but not discriminable. However, because of the geometry of information, activity can be present and not discriminable along discriminant line 2 in block A, but it must additionally be present in other dimensions that are orthogonal to discriminant line 2, which is our main point. There must be unique dimensions across blocks, and this is supported by our analyses.

## Learning dependent changes in information

We next examined changes in these subspaces, $w$, with learning (Figs 6 and 7). For the What blocks we examined the hypothesis that population activity patterns depended only on the chosen image. This would be a visual response that did not depend on value, and therefore did not change with learning. The x-phase analysis (Fig 4) has already characterized this in one way. To characterize changes around the reversal period, trial-by-trial, we analyzed changes across phases, from acquisition to reversal (Fig 6). As shown previously, acquisition choices projected into the acquisition subspace (Fig 6A Red and Blue) and reversal choices projected into the reversal subspace (Fig 6B Orange and Cyan), for an example What block, were well separated in their corresponding subspaces. However, when we projected reversal trials into the acquisition space (Fig 6A Orange and Cyan) or acquisition trials into the reversal space

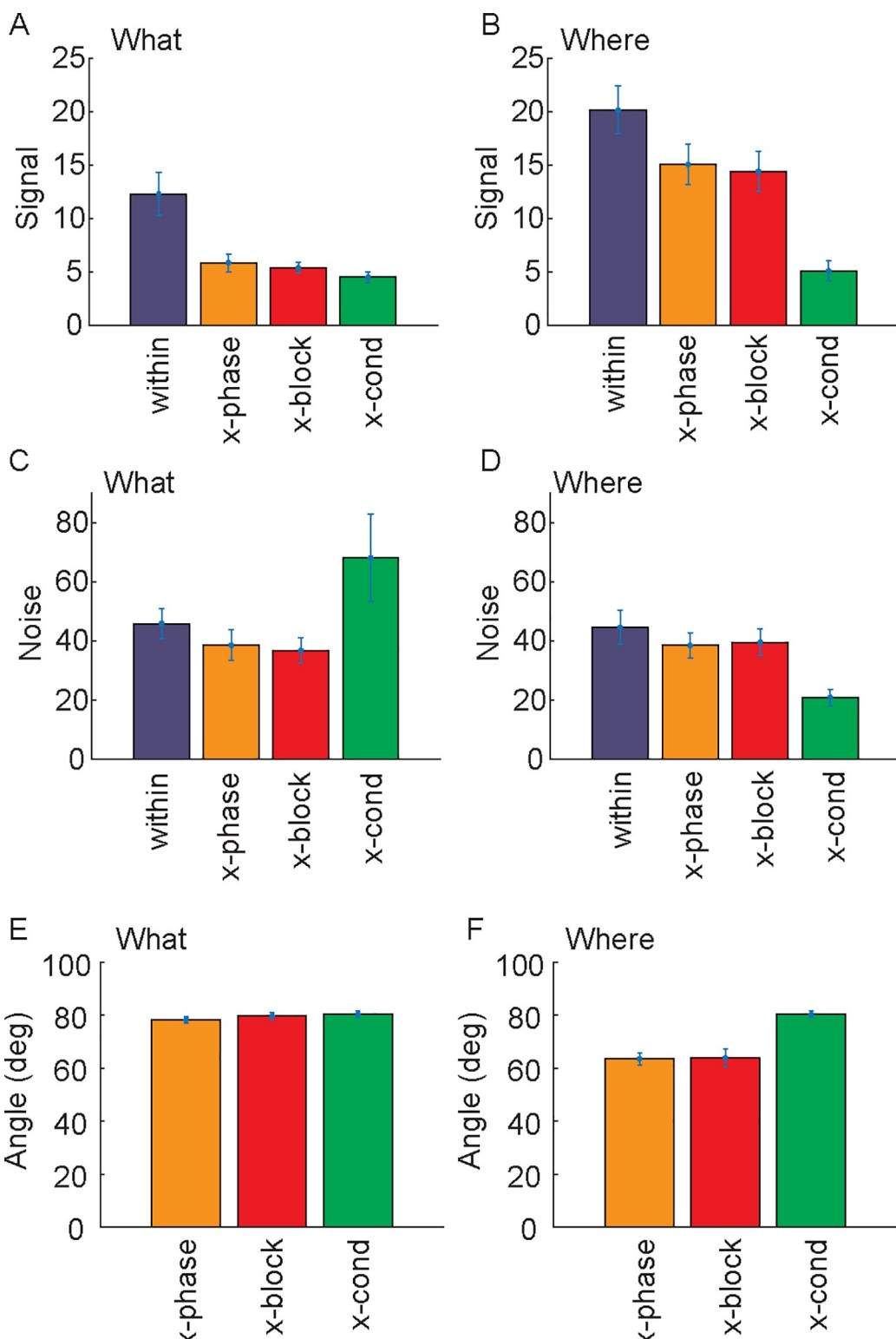

**Fig 5. Signal, noise and principal angle.** A. Signal (which for linear information is the norm of the difference in the mean responses in the two conditions, i.e. the norm of $w = \mu_2 - \mu_1$) for the What condition, after projecting data into each subspace. The subspace for a single block is given by activity in the two time bins (i.e. 0–250 ms after cue onset and 251–500 ms after cue onset). B. Same as A for the Where condition. C Noise (Trace of the noise covariance matrix, after projecting data into corresponding subspace) for the What condition. D. Same as C for the Where condition. E. Principal

angles for the What condition. This is the angle between the subspace for the current block and the opposite phase of the current block (x-phase), other blocks of the same type (x-block) and other blocks of the other type (x-cond). Note the principal angle is given by the matrix norm of the matrix of dot products between all dimensions of each subspace. F. Principal angles for the Where condition. The principal angle is larger between subspaces for different blocks of the What condition than the where condition.

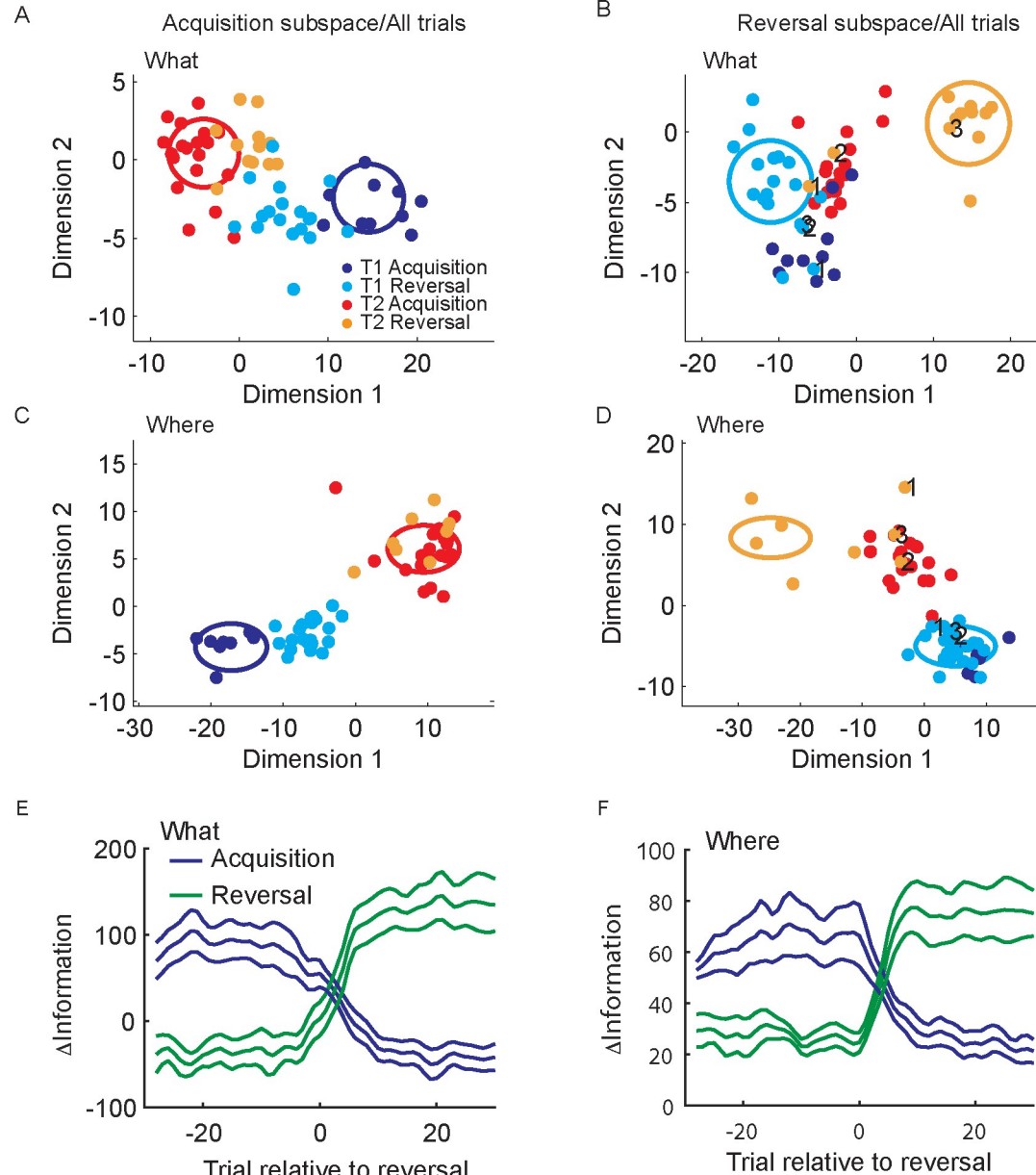

**Fig 6. Population activity in acquisition and reversal subspaces.** A. Projection of single trial population activity from the acquisition and reversal phases into subspace defined for the acquisition block for an example What block. Each dot is a trial. T1 is target 1 and T2 is target 2. Dimensions are rotations within the subspace spanned by the two time bins, from 0–250 and 251–500 ms after cue onset. The dimensions were found by computing the eigenvectors for the covariance across these two time bins. B. Projection of single trial population activity from both phases into the subspace identified for the reversal phase. Dot color indicates phase and object chosen. Numbers indicate trials after reversal: 1 is first trial after, 2 is second trial, etc. C. Same as A for an example Where block. D. Same as B for an example Where block. E. Average difference in distances in subspaces identified for acquisition and reversal for What blocks. F. Same as E for Where blocks. Error bars are s.e.m. N = 6.

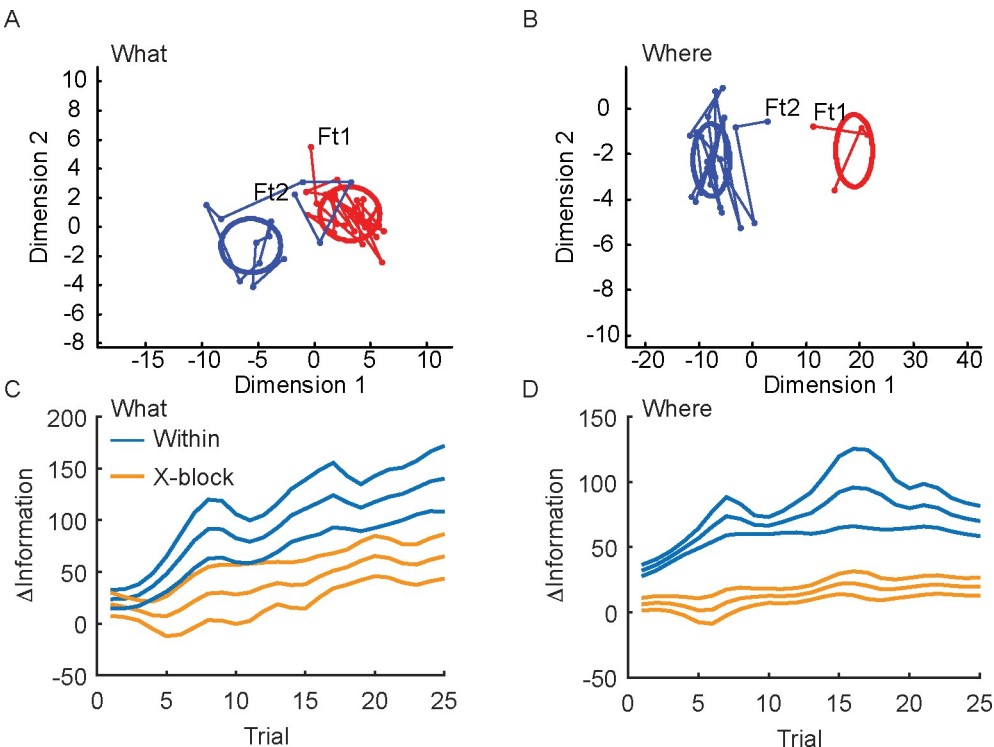

**Fig 7. Convergence of population activity with learning.** A. Example block of trials in which the population activity initially starts far from the distribution of activity after learning. Activity evolves and converges to the post-learning distribution. Ft1 is the first trial for object 1 and Ft2 is the first trial for object 2. Additional linked points are subsequent trials of the same choice, which are not necessarily consecutive trials in the task. Because the number of times each option was chosen in each block varies, the number of points varies. Option 1 was chosen more often than option 2 in this example block. Ellipses show 1 standard deviation of data for each condition. Dimensions 1 and 2 on the x and y axes refer to the subspace for each block, which is defined by rotations within the subspace spanned by the activity in the two time bins (i.e. 0–250 and 251–500 ms after cue onset). B. Same as A for an example Where block. C. Separation of population activity patterns with learning, in both the within and x-block subspaces for the What blocks. Y-axis indicates the difference between the Mahalanobis distances of the single trial activity to the mean for the opposite vs. same condition (see methods). Larger distances indicate further from the opposite choice distribution and closer to the correct choice distribution. D. Same as C for the Where blocks. Note, values for x-block ΔInformation are low because we did not re-estimate means after projection for this analysis, as we did for the analyses in Fig 4. Error bars are s.e.m. with n = 6.

(Fig 6B Blue and Red) they were less well separated. This could also be seen in an example Where block (Fig 6C and 6D). In addition, it can be seen in the example Where block, that the reversal trials remain linearly separable in the Acquisition subspace (Fig 6C, Cyan and Orange dots), even though they move closer together. This is consistent with a decrease of information without a decrease in decoding accuracy since information is the distance between the clouds and accuracy is the fraction of points that can be linearly separated. Furthermore, in the reversal phase, the first few trials following the reversal have activity patterns consistent with the acquisition phase (Fig 6B and 6D, numbered points for each choice). Thus, it takes a few trials for the animals to infer the reversal, and during this period the population activity patterns are more consistent with the acquisition phase [15]. When we examined this effect on average, we found, consistent with initial learning, that reversing the choice-outcome mapping drove activity into a new subspace for What blocks (Trial x Subspace; $F(58, 290) = 17.4$, $p < 0.001$) and Where blocks (Trial x Subspace; $F(58, 290) = 16.5$, $p < 0.001$).

We further characterized changes during initial learning. If the activity only reflected the visual attributes of the chosen image, then the population activity during initial learning should

be the same as the population activity after learning. The other possibility is that the subspace changed as the values of the images were learned. In this case, population activity will initially differ from the population activity after learning, converging to the learned distribution as values are learned. To characterize this we projected the population activity, trial-by-trial, into the low dimensional space *w* specific to the current block (within) as well as other blocks of the same condition (x-block) for comparison. We then compared the activity early in learning to the activity later, as the choice values were learned. Activity was compared by calculating the difference in Mahalanobis distances between single trials and the centroid of the distribution for the opposite and chosen options in each trial (see methods). As neural activity in single trials approached the learned distribution for the chosen option and moved away from the distribution for the unchosen option this distance should increase. We found that the population activity did not distinguish the chosen options when they were initially chosen (Fig 7A and 7C). When we projected the population activity, trial-by-trial, onto the low dimensional space relevant to decoding, we could see that the activity patterns tended to be similar for choices of the two options in the early trials (Fig 7A, Ft1 and Ft2). However, as the animals gained experience with the options, and they learned which one was more valuable, the activity patterns became more differentiated, and converged to the stable distribution seen following learning. When we examined this on average, we found that the activity patterns in the first few trials were more similar for the two choices. However, with learning the patterns became more differentiated (Fig 7C). When we compared the relative information within the block's subspace to the relative information in subspace from other blocks of the same type, the distances diverged with learning (Trial x Subspace; $F(24, 120) = 3.7$, $p < 0.001$). We also found that these distances diverged in the Where blocks with learning (Fig 7B and 7D; Trial x Subspace; $F(24, 120) = 2.1$, $p = 0.005$). When we examined the first trials, we found that they were different in the Where blocks ($F(1, 5) = 53.4$, $p = 0.001$), but not in the What blocks ($F(1,5) = 0.0$, $p = 0.955$). Thus, the neural subspaces in dlPFC did not simply reflect choice of an object or choice of a direction. They changed with learning and therefore reflected the learned values associated with these choices. There was, however, more overlap in the subspace for learned directions than learned objects, again consistent with the dimensionality of the spaces and the preserved information and decoding performance across blocks in the Where condition, relative to the What condition.

## Dimensionality of reward related activity

In a final series of analyses, we examined the dimensionality of the neural activity at the time of reward delivery (Fig 8). We again used two 250 ms bins, but this time locked to the time of reward (or no reward). In these analyses we defined the mean activity vector $\mu_m$ for the simultaneously recorded population for rewarded and unrewarded trials (i.e. $m = 1$ for reward and $m = 2$ for no reward), instead of direction or chosen image related activity. When we compared the dimensionality of the neural responses to reward in What and Where blocks, we found that there were no differences between conditions (Fig 8A; $F(1, 6) = 2.0$, $p = 0.216$). When we examined the dimensionality of the difference between rewarded and non-rewarded trials, $w = \mu_{m = reward} - \mu_{m = noreward}$, there was also no difference between the What and Where conditions (Fig 8C; $F(1, 6) = 2.7$, $p = 0.162$). There were also no differences when we examined the number of dimensions required to account for 80% of the variance in the average activity (Fig 8B; $F(1, 6) = 2.6$, $p = 0.169$)) or the differences, $w$ (Fig 8D; $F(1, 6) = 1.7$, $p = 0.252$). Therefore, there were no differences in the dimensionality of the neural responses that encoded reward outcome between What and Where blocks.

When we examined information and fraction correct for rewarded vs. non-rewarded outcomes, we found that the Information did not vary across subspaces (i.e. comparing within, x-

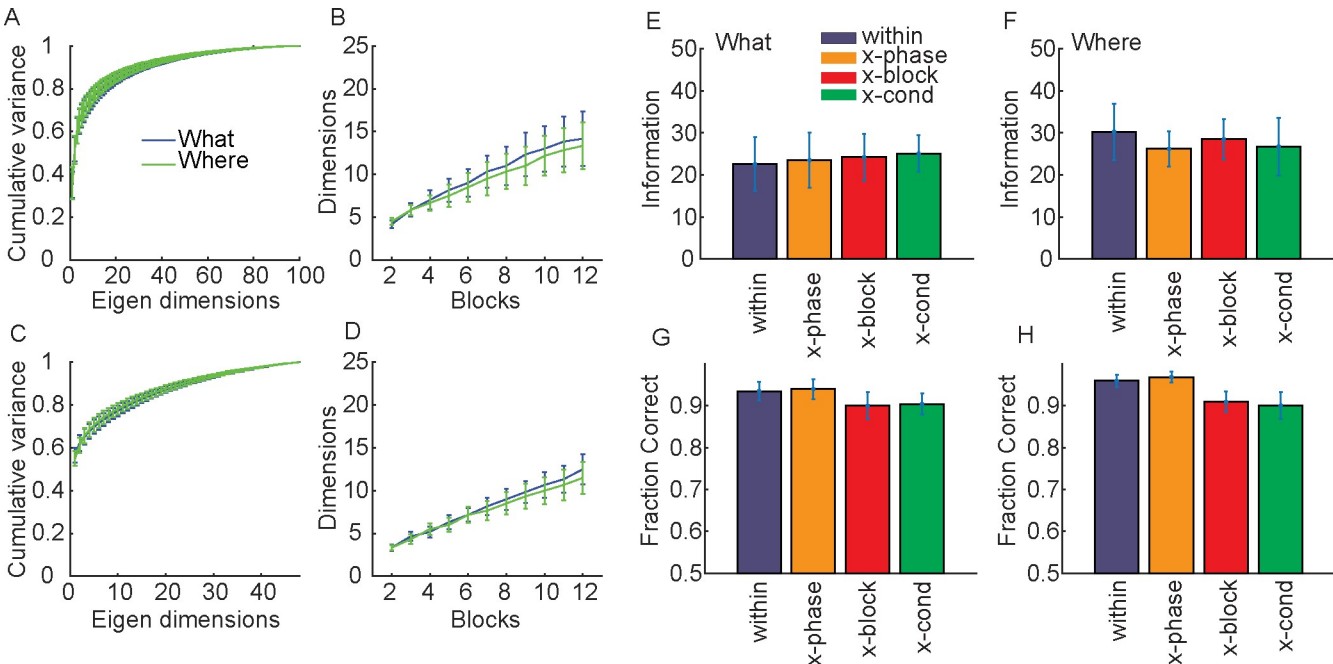

**Fig 8. Dimensionality and information in reward related activity.** A. Cumulative fraction of variance explained in trial-averaged responses to the reward (i.e. the vectors $\mu_i$ from each block and phase where $\mu_1$ is the activity for reward and $\mu_2$ is the activity for no reward) across all blocks split out by condition. B. Number of eigen dimensions necessary to account for 80% of the variance as a function of the number of blocks included in the analysis. C. Same as A for informative dimensions, $w = \mu_2 - \mu_1$, where again $\mu_1$ is the activity for reward and $\mu_2$ is the activity for no reward. D. Same as B for informative dimensions. E. Information about reward delivery in the What condition. F. Information about reward delivery in the Where condition. G. Fraction correct for the What condition, where prediction is whether a reward was or was not delivered. H. Same as G for the Where condition.

phase, x-block and x-cond) for the What condition (Fig 8E; F(3, 5) = 0, p = 1.000) or the Where condition (Fig 8F; F(3, 5) = 0.8, p = 0.509). There were differences in fraction correct across subspaces for the What condition (Fig 8G; F(3, 5) = 4.1, p = 0.026) and the Where condition (Fig 8H; F(3, 5) = 4.8, p = 0.016). Thus, there tended to be no variation in information about reward outcome across subspaces, although there was a small variation in fraction correct.

## Discussion

The neural code for the value of choices in dlPFC exists in a high dimensional space, where the maximum number of dimensions is given by the total number of neurons. We recorded from large populations of these neurons using 8 Utah arrays, with 4 implanted in each hemisphere. Although the total number of possible activity patterns in our recorded populations is very large, the number of patterns generated in any given experiment is only a small fraction of this. When we estimated the number of dimensions visited in our experiment, which is a measure of the total diversity of population patterns, we found that the dimensionality of neural activity across a session was higher for learning the values of novel pairs objects than it was for learning the values across a pair of directions, in dlPFC. When we examined this block-by-block, we found that object value, for different pairs of objects, was represented in different dimensions or regions of dlPFC coding space. Because of the high dimensionality of prefrontal population codes, this led to reduced interference or cross-talk about object value between blocks. Therefore, when the values for a new pair of objects was learned, they did not interfere with the learned values from other blocks. When values were learned for new pairs of objects,

the brain can use new regions of coding space to segregate those value from the values of other pairs of objects. On the other hand, when the animals learned action values, for actions that were repeated across blocks, activity for action value in different blocks re-occurred in the same regions of coding space. When we examined the changes of action and object representations with learning, we found that it was not only the visual image and motor kinematic properties that drove representations into novel coding regions. Prefrontal cortex population representations did not distinguish choices of the two images when they were first chosen. The representation began to distinguish choices of the images as their values were learned. This was also true, although to a smaller extent, of action values. Therefore, dlPFC population representations do not simply reflect visual stimulus or motor properties of choices, they also reflect the values of the objects or actions.

The recent advent of technologies for very high channel count recordings (HCRs) has driven increased interest in understanding how populations of neurons code information [10, 28]. Dimensionality reduction techniques have been used to understand HCR data [2, 5, 29]. These techniques allow one to reduce the dimensionality of datasets, such that neural activity can be visualized in 2 or 3 dimensions instead of the 100s of dimensions present in the original data. When the dimensionality is reduced, however, variance must be thrown away. This raises the question of whether something is being missed when the dimensionality is reduced. Surprisingly, it is often the case that relatively low dimensional approximations can capture much of the variance in high dimensional datasets [2, 3, 23, 24, 30, 31]. This shows that neural activity in populations of neurons resides on relatively low dimensional manifolds. Stated another way, only a subset of the possible patterns of activity that can theoretically be produced by a population are produced.

Theoretical work has shown that the low dimensional nature of many datasets, particularly from experiments in which animals are executing simplified tasks, arises because of the limited number of task conditions[6]. When there are few conditions, neural activity cannot be driven to explore high dimensional space, and therefore the responses of even a large population of neurons can be approximated using a relatively small number of dimensions. This also accounts for previous findings that a few neurons can account for the behavior of animals relatively well [32]. Why do we need neural populations if this is the case? The answer (at least one of the answers) is that, when tasks become more complex, the dimensionality of the neural representation underlying the task becomes higher, and a larger number of neurons is required to span the increased dimensionality [6, 23]. Much of the low-dimensional nature of neural representations in laboratory experiments comes from the restricted conditions. In natural behaviors neural activity would necessarily explore higher dimensional space.

We found that dimensionality was higher across blocks of the What condition than across blocks of the Where condition. The dimensionality is a function of both the number of conditions used in the task, and, the difference in the pattern of population activity across the conditions [6]. If we had used different directions across blocks, the dimensionality of the Where condition would likely have been higher. However, there would be a limit to the dimensionality increase possible with directions. This is because direction of eye movements in cortex is encoded by relatively smooth Gaussian functions. Thus, two similar directions would lead to similar patterns of population activity, and therefore they could be represented with similar dimensions in coding space. Objects, on the other hand, occupy a much larger dimensional space. Had we used, for example, oriented Gabor patches instead of images, we would likely have found a lower-dimensional space, similar to what was seen with directions [33].

Many of the previous studies on dimensionality have focused on temporal dynamics, whereas we have focused mostly on static population codes. However, temporal dynamics and the complexity of the code for decisions, drive dimensionality in a related way, at least as they

are normally analyzed [6, 34]. Temporal dynamics function as another task dimension. Therefore, if we included temporal dynamics (beyond the two time points we used) we would have arrived at a similar answer with respect to learning driven increases in dimensionality. Although we explored this option, we did not have enough trials in each block to accurately estimate both temporal dynamics and coding of object choice.

Other studies have examined neural population dimensionality in other ways. For example, in V1, the dimensionality of neural representations is maximal when the system is driven using white noise stimuli, which have no spatial or temporal structure [33]. In addition, Cowley et al. found that gratings and natural movies drove population responses within a subset of the dimensions driven by the white noise stimuli. Another study compared dimensionality in V1 and motor cortex and found that the dimensionality was primarily constrained in motor cortex by temporal correlations among neurons, which reflect the intrinsic dynamics of the motor system [4, 35]. In visual cortex dimensionality was primarily constrained by shared tuning properties of the neurons. Therefore, the way in which dimensionality of neural population activity is constrained may reflect the underlying computational role of the population, and likely will not be a generic property of neural populations. Our study shows that dimensionality of neural activity in dlPFC is a function of the actions and objects about which the animals is learning, and the learned values of those options.

Other work has examined learning of population patterns of activity, in the context of a closed-loop brain machine interface experiment [36]. This study found that it was easier for animals to learn to control a BMI cursor, when the mapping between neural activity and cursor motion required the generation of population patterns of activity that were within the manifold that characterized activity during normal behavior. When the animal was required to generate patterns of activity off this manifold to drive the cursor, learning was more difficult. Recent follow up analyses have further characterized this result by showing that the learning primarily drove re-association of activity patterns present before learning to new movements [26], although additional experiments have shown that new patterns can be generated with additional training [27]. Thus, motor cortex can generate new patterns of activity, but only after extensive training. This result, in combination with our result, raises the question of whether it would be easier for some brain areas, for example dlPFC, to learn to generate novel patterns of activity. Or, whether the dimensionality expansion we have seen in our experiment is related to the fact that we have only explored learning of a relatively small set of objects relative to what one can learn in life. It is possible that the dimensionality would eventually saturate. If we assume that Hopfield networks are an approximate model of dlPFC learning, a capacity limit would eventually be reached [37]. Whether the dlPFC manifold is higher dimensional than the motor cortical manifold remains an open question. It is also not clear from these experiments whether we are driving neural activity into regions of coding space that have never been previously visited. However, it is possible because the animals have never seen these images before, and they have never learned the values of these images. Learning is likely to be important for driving novel patterns of neural activity.

Recently it has been shown that dlPFC neurons show mixed selectivity [38], and that this selectivity can develop with Hebbian learning[39]. Whether Hebbian mechanisms are engaged by RL is currently unclear. Mixed selectivity gives dlPFC neural populations a flexible representation of task relevant variables, such that they can be decoded or read out in arbitrary ways. Thus, one can build a linear decoder to discriminate arbitrary associations between stimuli and actions. These arbitrary associations represent cognitive rules, for example, green means go and red means stop, and not values. Although, if one is unable to learn the correct cognitive rules there will be clear value implications. These flexible representations arise because of the non-linear mixing of the representation of task variables in neural activity. The

nonlinear mixing effects expand the dimensionality of the neural representation relative to strictly linear encoding. However, the effects are due to fixed nonlinear interactions between task factors within single trials, whereas our effects accumulate across blocks and do not occur within single trials. Furthermore, mixed-selectivity addresses a different question than we are addressing. Mixed selective representations allow for linear decoding of arbitrary combinations of fixed, over-learned task conditions, which gives these codes their flexibility. Whereas we are examining the effects of learning on increasing the dimensionality, or the number of regions visited by neural activity, of dlPFC representations.

## PFC representations and reinforcement learning

Most studies of learning have focused on how we learn a single decision between a pair of objects in bandit tasks [40, 41], or in Pavlovian paradigms the state value of a stimulus [42]. In real life, however, we must learn and track the values of a large number of objects and actions. While visual cortex can generate population representations that distinguish among large sets of objects [43], it has not been clear how values can be learned across large sets of objects. Here we show that when values for pairs of objects are learned, these representations form in regions of dlPFC coding spacing that are relatively independent of the regions used to represent other pairs of objects. This provides an easy way to distinguish values for novel pairs of objects, since the values for the different pairs can be read-out using different decoding models. This is because of the mapping between novel regions of coding space and the corresponding decoding models. Thus, dlPFC population codes make efficient representations.

The learned representations in dlPFC were not a simple consequence of the visual stimuli used or the motor response required, as might be the case in sensory or motor areas. The representations developed with learning, and changed when the values of the choices were reversed [15]. Thus, learning drove neural activity into parts of population space that would not be explored under passive visual experience or motor activity, because passive visual experience and motor activity did not change across learning, but the neural representation did. There was also some evidence that reversing stimulus outcome or action outcome mappings drove activity away from the subspace used for the opposite mapping for action learning. This suggests an active process, where reversing values on actions drove activity into regions of coding space that were more different than relearning the values of actions in a different block.

The exact role of the dlPFC in learning in this task is not currently clear. We have previously shown that lesions to the ventral-striatum causes deficits specific to the What condition in this task, with no effects on the Where condition [17]. We have also shown, in other work, that dopamine antagonist injections into the dorsal striatum, can disrupt action learning, likely consistent with our Where condition [44]. Because the dlPFC has strong anatomical connections with the dorsal striatum and not the ventral striatum, this would suggest that the representations we see in dlPFC may be more critical for Where learning than What learning [45]. However, it is still unclear how cortex and striatum interact to drive learning in these paradigms. Much work emphasizes a specific role for cortical circuits across broad classes of learning problems [46]. Additional work will be required to understand how diverse frontal-striatal circuits, and their interaction with the thalamus, amygdala and other structures underlie different forms of learning.

## Conclusion

Overall, our data support the conclusion that learning to choose between good and bad objects or actions drives dimensionality expansion in dlPFC. Although these signals exist in dlPFC, they likely exist elsewhere too. We did find, however, that learning the values of new pairs of

objects drives novel patterns of activity in dlPFC populations, that exist in subspaces relatively independent of those generated for other pairs of objects. Furthermore, these subspaces change with learning, such that passive experience with the objects or actions does not generate these patterns of activity. In addition, reversing the values of the objects also drove activity into novel locations in dlPFC coding space. Finally, learning and relearning reward values for a preserved set of directions reuses similar locations in dlPFC coding space. Overall, this study shows that learning is a dynamic process, and learning the difference between good and bad choices can drive novel patterns of neural activity in dlPFC.

## Methods

### Ethics statement

All experimental procedures were performed in accordance with the ILAR Guide for the Care and Use of Laboratory Animals and were approved by the Animal Care and Use Committee of the National Institute of Mental Health. Two male monkeys (*Macaca mulatta*, *W*—6.7kg, age 4.5yo, *V*—7.3kg, age 5yo) were used as subjects in this study. All analyses were performed using custom made scripts for MATLAB (The Mathworks, Inc.).

### Task

Each block consisted of 80 trials and one reversal of the stimulus based or action based reward contingencies (Fig 1A). On each trial, monkeys had to acquire and hold a central fixation point for a random interval (400–600 ms). After the monkeys acquired and held central fixation, two images appeared, one each to the left and right (6˚ visual angle from fixation) of the central fixation point. The presentation of the two images signaled to the monkeys to make their choice. The monkeys reported their choices by making a saccade to their option, which could be based on the image or the direction of their saccade. After holding their choice for 500 ms, a reward was stochastically delivered. In What blocks one of the images was rewarded 70% of the time and the other 30%, and in Where blocks one of the directions was rewarded 70% of the time and the other 30%. If the monkeys failed to acquire central fixation within 5 s, hold central fixation for the required time, or make a choice within 1 s, the trial was aborted and they repeated the trial.

Each block used two novel images. The images were randomly assigned to the left or right of the fixation point for every trial. The images were changed across blocks but remained constant within a block. What and Where blocks were randomly interleaved throughout the session. For What blocks, reward probabilities were assigned to each image, independent of the saccade direction necessary to select an image. Conversely, for Where blocks, reward probabilities were assigned to each saccade direction, independent of the image at each location. The block type (What or Where) was held constant for each 80-trial block. There were 12 blocks of each condition in each recording session. The choice-outcome mapping was reversed on a randomly chosen trial between 30 and 50, inclusive. The reversal trial was independent of the monkey's performance and was not signaled to the monkey. At the reversal in a What block, the less frequently rewarded image became the more frequently rewarded image, and vice versa. At the reversal in Where blocks, the less frequently rewarded saccade direction became the more frequently rewarded saccade direction, and vice versa.

Images provided as choice options were normalized for luminance and spatial frequency using the SHINE toolbox for MATLAB [47]. All images were converted to grayscale and subjected to a 2-D FFT to control spatial frequency. To obtain a goal amplitude spectrum, the amplitude at each spatial frequency was summed across the two image dimensions and then averaged across images. Next, all images were normalized to have this amplitude spectrum.

Using luminance histogram matching, we normalized the luminance histogram of each color channel in each image so it matched the mean luminance histogram of the corresponding color channel, averaged across all images. Spatial frequency normalization always preceded the luminance histogram matching. Each day before the monkeys began the task, we manually screened each image to verify its integrity. Any image that was unrecognizable after processing was replaced with an image that remained recognizable.

Eye movements were monitored and the image presentation was controlled by PC computers running the MonkeyLogic (version 1.1) toolbox for MATLAB [48] and Arrington Viewpoint eye-tracking system (Arrington Research, Scottsdale, AZ).

## Data acquisition and preprocessing

Microelectrode arrays (BlackRock Microsystems, Salt Lake City, USA) were surgically implanted over the prefrontal cortex (PFC), surrounding the principal sulcus (Fig 1B). Four 96-electrode (10×10 layout) arrays were implanted on each hemisphere. Details of the surgery and implant design have been described previously (Mitz et al., 2017). Briefly, a single bone flap was temporarily removed from the skull to expose the PFC, then the *dura mater* was cut open to insert the electrode arrays into the cortical parenchyma. The dura mater was then sutured and the bone flap was placed back into place and attached with absorbable suture, thus protecting the brain and the implanted arrays. In parallel, a custom designed connector holder, 3D-printed using biocompatible material, was implanted onto the posterior portion of the skull.

Recordings were made using the Grapevine System (Ripple, Salt Lake City, USA). Two Neural Interface Processors (NIPs) made up the recording system, one NIP (384 channels each) was connected to the 4 multielectrode arrays of each hemisphere. Behavioral codes from MonkeyLogic and eye tracking signals were split and sent to each NIP. The raw extracellular signal was high-pass filtered (1kHz cutoff) and digitized (30kHz) to acquire single unit activity. Spikes were detected online and the waveforms (snippets) were stored using the Trellis package (Grapevine). Single units were manually sorted offline. Data was collected in 6 recording sessions (3 sessions per animal).

## Subspace identification

Our analyses of population activity began by identifying informative subspaces. Informative dimensions (equivalent to linear discriminant lines generated with Fisher Discriminant Analysis) are given by

$$w = Q^{-1}\Delta\mu$$

Where $Q$ is the noise covariance matrix, and $\Delta\mu = \mu_2 - \mu_1$ is the vector of differences in mean responses for the two conditions[7, 49]. The vector $\mu_i$ is given by the average response of the population for choice $i\epsilon\{1,2\}$. In What blocks the choices, $i$, corresponded to the two images and in Where blocks the choices corresponded to the two directions. The elements, $j$, of the vector are the average response, $x$, of each neuron $j$, where the expectation is taken across trials, $k$, to choice $i : \mu_{i,j} = 1/K\sum_{k=1}^{K} x_{i,j,k}$. Because we did not have a large number of trials to estimate the model, we approximated the covariance matrix, $Q$, by the identity matrix. We and several others have shown in the past that diagonal approximations to $Q$ often perform as well as the full $Q$, and other groups working with very large numbers of neurons are taking the same approach[25]. Assuming an identity matrix is equivalent to carrying out nearest centroid classification[50]. As shown in the results, decoding performance was high, so this approximation worked well. We tried several regularized logistic regression approaches, but found that using the difference in means to define $w$ gave the best performance. Therefore, for our analyses we used $w = \Delta\mu$.

We defined linear discriminant lines, *w*, for each phase (i.e. acquisition and reversal) of each block, and we also estimated *w*'s for two time bins (0–250 and 251–500 ms after cue onset) in each block and phase. Therefore, the informative subspace for a single block and phase was two dimensional—one dimension for each time bin. These initial informative dimensions were identified using leave-3-out cross validation. Therefore, the dimension, *w* was estimated separately for each trial to be tested, by first leaving out the current trial, and the preceding and following trial, when the averages, $\mu_{i,j}$, were calculated.

In some blocks, when the animal rapidly discovered the correct option and chose it almost exclusively, insufficient trials were available to estimate the model, because we needed sufficient trials of each choice. These blocks were not analyzed. We also removed the first 4 trials of each block and treated these as learning trials. This was because the animals' choice accuracy exceeded chance ($p < 0.05$) on trial 4 of the What condition and on trial 6 of the Where condition. Therefore, the informative subspaces were defined on the neural activity after learning and during performance.

## Dimensionality and variance

To estimate dimensionality in each condition, we carried out singular value decomposition on the matrix of linear discriminant lines, *w* accumulated across blocks and phases, or on the matrix of means, $\mu_i$ as indicated in the results. For estimates of dimensionality the linear discriminant lines were averaged across the cross validated trials for each block and phase. We collected these into a matrix defined by

$$W = [w_{1,1,1} \ldots w_{l,m,n} \ldots w_{24,2,2}]$$

Where the subscript *l* indicates block ($1 \leq l \leq 24$), *m* indicates the acquisition or reversal phase ($1 \leq m \leq 2$) and *n* indicates time bin ($1 \leq n \leq 2$). The row dimensionality of this matrix was given by the number of neurons in each session, *S*, and the column dimensionality by the number of blocks, phases and time bins (*l x m x n* = 96). In all analyses we had more neurons than discriminant lines (96 for the whole task, 48 for each condition type), so the conditions and not the neurons constrained the rank of the matrix. To estimate the variance accounted for as a function of dimensions for the whole experiment, we did an eigenvector decomposition of $WW^T$. In a second analysis this was estimated separately for each condition. For this we split *W*, putting the What blocks (n = 12) into one matrix and the Where blocks (n = 12) into the other and carrying out the eigen decomposition on each matrix separately.

## Decoding performance

When we carried out the decoding analysis, we first projected all data into the indicated subspace, preserving the cross validation. For example, when we carried out the decoding analysis 'x-phase', we projected the data from the acquisition phase of the block, trial-by-trial, into the reversal phase subspace of the same block, or vice-versa. We therefore projected each individual trial, $x_{i,k}$, into the indicated subspace $W_{l,m}$:

$$z_{i,k} = W_{l,m}^T x_{i,k},$$

Where the matrix $W_{l,m} = [w_{l,m,1}, w_{l,m,2}]$ has two columns, one for each time bin, and *l* and *m* indicate the block and phase, as above. After projection of the data into this subspace, we carried out leave-one-out cross validated linear decoding by re-estimating decision boundaries in the new subspace. This was done by computing means within the new subspace, $\mu_{i,j}{}^z =$

$K^{-1} \sum_{k=1}^{K} z_{i,j,k}$ and then calculating the Mahalanobis distance between each trial and the

corresponding mean for each condition,

$$M_i = (z_{i,k} - u_i^z)^T Q^{-1} (z_{i,k} - u_i^z).$$

For these distance measures we used cross validated pooled covariance matrices, $Q$, estimated using leave-one-block out cross validation. Therefore, these distances were calculated using covariances estimated from all trials not including the trials from the current block. Similar answers were obtained when Q was replaced by the identity matrix. The trial was then classified to the category, $i$, to which it had the smallest distance, $M_i$. Therefore, if the neural activity only changed location within the same subspace across conditions, classification performance would not change.

## Information measure

We computed information after projecting the data into the informative subspaces, as described for the decoding analyses. To simplify, for the information measure, we stacked the relevant *w's* for the two time bins into one long vector, i.e. $w = \begin{bmatrix} w(t) \\ w(t+1) \end{bmatrix}$, and also stacked the activity for the two time bins into another vector e.g. $x = \begin{bmatrix} x(t) \\ x(t+1) \end{bmatrix}$, and then carried out the projection. After projecting onto this stacked *w*, information can be calculated with:

$$I = \frac{(\mu_2 - \mu_1)^2}{\sigma^2}.$$

The means, $\mu_i$ and variance, $\sigma^2$ were estimated after the data was projected into the subspace. After projection, $\mu_i$ is a scalar, and $\sigma^2$ is the variance of the scalar distribution. We can estimate the fraction of errors[7], i.e. the fraction of times the model (m) and the animal (c) did not choose the same option,

$$p(m = 2|c = 1) = (2\pi)^{-1/2} \int_{\sqrt{I}/2}^{\infty} \exp\left(\frac{-y^2}{2}\right) dy.$$

We compared this estimate, based upon our information calculation, to the results of carrying out cross validated decoding on the data after projecting onto *w* (Fig 4D). For an arbitrary projection, $w_s$, the information is given by:

$$I = \frac{(w_s^T (\mu_2 - \mu_1))^2}{w_s^T Q w_s}.$$

Note that if we plug in the optimal linear estimator, $w = Q^{-1}\Delta\mu$, this is the equation for Fisher Information[7].

## Mahalanobis distance for learning analyses

We also calculated the trial-by-trial Mahalanobis distance during learning, to examine the evolution of neural representations. Specifically, if $z_{i,k}$ is the population response on trial $k$ in which the animal chose option i, after projection into the corresponding subspace, the Mahalanobis distance to the distribution for option i *is*

$$M_i = (z_{i,k} - u_i^z)^T Q^{-1} (z_{i,k} - u_i^z).$$

Correspondingly, the distance to the distribution for option $j$ is

$$M_j = (z_{i,k} - u_j^z)^T Q^{-1} (z_{i,k} - u_j^z).$$

The plotted values for learning in the results are $\Delta$information = $M_j - M_i$, i.e. the difference in the distances to the distributions for the two choices. As the activity nears the learned distribution for the correct response, $M_i \rightarrow 0$ and $M_j$ increases. Thus, $\Delta$information increases with learning, as shown.

### Single neuron ANOVA

We carried out an ANOVA analysis on the responses on each single neuron, in 50 ms bins time locked to cue onset. The data for each recording session was first split into trials of the What and Where condition, which were analyzed separately. In each single trial, spikes were counted in 50 ms bins. The set of bins at a fixed time relative to cue onset were then subject to a univariate ANOVA, with fixed factors of chosen image, chosen direction, the current value of the choice estimated with a Rescorla Wagner RL model [17], the current block, and whether a reward was delivered. Image was nested under block as they were not comparable across blocks. Value was entered as a continuous variable. Results are reported as fraction of significant neurons at $p < 0.05$. Note that larger time bins would lead to a larger fraction of the population being significant for each factor.

### Acknowledgments

To perform the analyses described in this paper we made use of the computational resources of the NIH/HPC Biowulf cluster (http://hpc.nih.gov).

### Author Contributions

**Conceptualization:** Ramon Bartolo, Bruno B. Averbeck.

**Data curation:** Ramon Bartolo, Bruno B. Averbeck.

**Formal analysis:** Ramon Bartolo, Bruno B. Averbeck.

**Funding acquisition:** Bruno B. Averbeck.

**Investigation:** Ramon Bartolo, Bruno B. Averbeck.

**Methodology:** Richard C. Saunders, Andrew R. Mitz.

**Project administration:** Bruno B. Averbeck.

**Resources:** Bruno B. Averbeck.

**Software:** Ramon Bartolo, Bruno B. Averbeck.

**Supervision:** Bruno B. Averbeck.

**Validation:** Bruno B. Averbeck.

**Visualization:** Bruno B. Averbeck.

**Writing – original draft:** Ramon Bartolo, Bruno B. Averbeck.

**Writing – review & editing:** Ramon Bartolo, Bruno B. Averbeck.

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
