## [Decision Letter · Decision Letter 0]

20 Dec 2019

Dear Dr Averbeck,

Thank you very much for submitting your manuscript, 'Dimensionality, information and learning in prefrontal cortex', to PLOS Computational Biology. As with all papers submitted to the journal, yours was fully evaluated by the PLOS Computational Biology editorial team, and in this case, by independent peer reviewers. The reviewers appreciated the attention to an important topic but identified some aspects of the manuscript that should be improved.

We would therefore like to ask you to modify the manuscript according to the review recommendations before we can consider your manuscript for acceptance. Your revisions should address the specific points made by each reviewer and we encourage you to respond to particular issues Please note while forming your response, if your article is accepted, you may have the opportunity to make the peer review history publicly available. The record will include editor decision letters (with reviews) and your responses to reviewer comments. If eligible, we will contact you to opt in or out.raised.

- Supporting Information uploaded as separate files, titled 'Dataset', 'Figure', 'Table', 'Text', 'Protocol', 'Audio', or 'Video'.

We hope to receive your revised manuscript within the next 30 days. If you anticipate any delay in its return, we ask that you let us know the expected resubmission date by email at ploscompbiol@plos.org.

Sincerely,

Samuel J. Gershman

Deputy Editor

PLOS Computational Biology

[LINK]

Reviewer's Responses to Questions

**Comments to the Authors:**

Reviewer #1: Bartolo and colleagues used simultaneous recordings of about a thousand neurons in the dlPFC of behaving monkeys to study learning dynamics of the neural representation of sensory input, action, and value. Surprisingly, even when the same task repeated (e.g., same rewarded action) the representation changed. Moreover, their results in Figures 6A-D portray similar dynamic flow of the neural representation in all cases studied, which is very intriguing.

Due to the pioneering nature of the study, many readers might not be very familiar with the analyses done by the authors. Partly, because the authors analyze different dimensions: one set of dimensions characterize the task/stimuli the other set the neural population. I think the authors could better explain the rational and meaning of each analysis as well as better define the analysis. Most of my specific comments below focus on that issue.

Specific comments:

1. Page 7: clarify: add neuron index to U=[mu…] – to emphasize you have many such vectors. Also (in the same sentence) does mean activity mean: trail average of spike count?

The matrix U is 192 times N, where N is the number of neurons. You should better motivate doing SVM on the 192 dimensions and not on the N dimensions, and explain its meaning (you are doing SVM on vectors that the ‘tuning curves’ of different neurons).

2. Figure 3D: the matrix W has only 96 columns. How can you show 100 eigen dimensions?

3. Fig 3C-H consider comparing with shuffled data (along the neuron dimension) of the same length.

4. Fig 3F: here the ‘principle component’ contributes about 40% for where and 20% for What, and the rest seem to add much less. It seems possible that the main difference between Where/What and Random in Fig 3F may result from this contribution of the first eigen dimension.

5. Fig S1 could be incorporated into the main body of the paper, per author’s judgement (same for S2 and S3).

6. Fig 4A – need to better explain (for example: what are the axes?)

7. Figure 4 C verses F – is it a ceiling effect?

8. Figure 4D: could you plot the what and where using different signs using the same color code of B,C,E,F?

9. Fig 4: did you look how information space/representation changes over to the second next block?

10. Page 11: please clarify what you mean by “different blocks of the What condition were more independent”?

11. Figure 5: what exactly are showing here? What is the ellipsoid? What are Dimenstion X? Why only four red points in Fig 5B? Also in Fig 5D – it seems counter intuitive that the DInformation for the X-block is around zero (orange traces?), whereas in Fig 4C&F information measures are very high.

12. Fig 6E&F: from Fig 4 one expects more information in where than in what – it seems that in Fig 6 this is opposite. Can you comment?

13. Results of Fig 6A-D are fascinating. If you will flip dimension 1 in 6C (multiply it by -1, I think it is perfectly fair) you will get almost the same plot as in 6A (also similar to some extent in 6D and 6B) – outlines a clear trend.

14. Page 18 typo {Oby, 2019 #6939}.

Reviewer #2: The authors studied population coding in NHP dlPFC during a context-dependent visual task. The task was blocked with two contexts: either the location of the stimulus was rewarded or the identity of the stimulus was rewarded, and in each case the reward was probabilistic (0.7 vs 0.3). Within blocks the authors reversed which location or which stimulus was reward, allowing for observation of choice reversal behavior. The data collected is rich and impressive due to the complex behavioral task and the large number of single units recorded using Utah arrays. The authors found that neural activity was high-dimensional and that neural activity explored different dimensions during different periods of the task. Such high-dimensionality has not been reported in previous studies because lower dimensional tasks were used (e.g. Rigotti et al 2013). Also, the finding that different dimensions are explored after value reversal is to my knowledge a novel finding. The paper is overall clearly presented but would benefit from some descriptive section headings in the results. Also, the paper lacks a few statistical controls that would help to support the claims in the paper.

Major (required to support claims/clarity)

1. There is no mention of cross-validation for the computation of dimensionality (Fig 3). One way to cross-validate is to separate the trials into 3 sets, compute the eigenvectors on the covariance between set1 and set2 and then project the eigenvectors onto set3 to get the variance of the eigenvectors in an unbiased way. Can you also justify why only 2 time bins were used?

2. Fig 3G,H make the claim that dimensionality increases over blocks, but there is no explanation of the null for this (dimensionality will increase as timepoints increase even for noise). I think with dimensionality computed in a cross-validated way this will be a less problematic analysis, but with un-cross-validated eigenvalues this analysis doesn’t tell us much.

3. Fig 4 again I am unsure what the control is for the F-tests. It would be nice for the reader to have a line on Fig 4B,C showing the chance amount of information for a randomly drawn vector from the subspace of responses.

4. Fig S2 needs more explanation in the figure/methods. By “each subspace” do you mean the single discriminant direction?

5. The paragraph after Fig 4 starts compellingly asking if responses depend on value, but this question is not answered until Fig 6. I am not sure what order these two figures should be in to best support the narrative, but personally I would have preferred Fig 6 before Fig 5.

6. Labels on Fig 5A,B axes as to what the dimensions are would be helpful to the reader. Why is Fig 5C labelled with change in information if it is a change in Mahalanobis distance? I think again it would be easier to understand after Fig 6.

Minor (may enhance paper but not required)

1. You are randomly presenting the reward, but the reason for this experimental design choice is not explained. Also, this reward period activity is not evaluated at all. Is this activity low-dimensional and independent of the task condition/phase? Does it code left/right or abstract quantities?

2. In the discussion there is mention that coding in new subspaces prevents interference with old subspaces and object-value associations. It would be interesting to see if the subspaces in Fig 6AC project differently onto reward-related directions (Fig 6A).

3. There is a mention of reversals in which the monkey quickly learned the reversal - is this due to a chance exploration of the other choice? Is there a difference in neural activity in these periods?

**Have all data underlying the figures and results presented in the manuscript been provided?**

Reviewer #1: None

Reviewer #2: No: Data is "available upon request"

PLOS authors have the option to publish the peer review history of their article (what does this mean?). If published, this will include your full peer review and any attached files.

Reviewer #1: No

Reviewer #2: No

---

## [Editor Report · Decision Letter 1]

11 Mar 2020

Dear Dr. Averbeck,

We are pleased to inform you that your manuscript 'Dimensionality, information and learning in prefrontal cortex' has been provisionally accepted for publication in PLOS Computational Biology.

Best regards,

Samuel J. Gershman

Deputy Editor

PLOS Computational Biology

---

## [Editor Report · Acceptance letter]

15 Apr 2020

PCOMPBIOL-D-19-01865R1 

Dimensionality, information and learning in prefrontal cortex

Dear Dr Averbeck,

I am pleased to inform you that your manuscript has been formally accepted for publication in PLOS Computational Biology. Your manuscript is now with our production department and you will be notified of the publication date in due course.

With kind regards,

Laura Mallard
